# TOGA: Trigger Optimization for Clean Data Ordering Backdoor Attack

## Abstract

Recent work has shown that backdoors can be learned in neural networks purely through the malicious reordering of clean training data, without modifying labels or inputs. These data ordering attacks rely on gradient alignment, ordering clean samples to approximate the gradients of an adversarial task. However, the effectiveness of such attacks depends greatly on the choice of the backdoor trigger, which determines how closely clean gradients align with the backdoor gradients. In this work, we introduce the first framework (**TOGA** - Trigger Optimization for Gradient Alignment) for optimizing triggers specifically for data ordering attacks. Our method significantly improves attack success rates by an average of 46% over prior methods across benchmark datasets (CIFAR-10, CelebA, and ImageNet) and sensitive application domains (ISIC 2018 for dermatology and UCI Credit-g for credit scoring), without compromising clean performance. We further show that optimized triggers can be adapted to create subpopulation-specific backdoors, selectively targeting vulnerable subpopulations. Finally, we show our method is robust against purification and gradient-similarity defenses. Our findings reveal new security and fairness risks for high-stakes domains, underscoring the need for broader defenses against data ordering attacks.

## 1 Introduction

Backdoor attacks manipulate neural networks to produce malicious outputs when a specific trigger is present in the input, while maintaining normal behavior on clean data (Gu et al., 2017; Li et al., 2022). These attacks often rely on data poisoning, where the attacker perturbs training inputs or labels to induce backdoor behavior (Chen et al., 2017; Zhong et al., 2020; Li et al., 2021b; Gao et al., 2024). Prior work has explored a wide variety of backdoor strategies, including imperceptible backdoors (Chen et al., 2017; Zhong et al., 2020; Li et al., 2021b; Gao et al., 2024), physical backdoors (Li et al., 2021a; Xue et al., 2021; Gong et al., 2023), and optimized backdoors (Doan et al., 2021b; Li et al., 2020; Zhang et al., 2023; Sun et al., 2024; Doan et al., 2021a), demonstrating the prevalence and versatility of this threat.

An emerging but understudied variant is the **data ordering attack**, where backdoors are introduced by simply *reordering* training instances while changing nothing else about the training data or procedure (Li et al., 2022; Shumailov et al., 2021). Shumailov et al. (2021) shows that reordering data batches can cause models to predict a backdoor class in the presence of a manually chosen trigger. This attack is particularly concerning because fixed seeds are commonly used for reproducibility in machine learning pipelines (Bethard, 2022; Dutta et al., 2022). With knowledge of the seed, an attacker can adversarially reorder data samples. The settings for data ordering attacks are increasingly plausible in large interdisciplinary teams (Krause-Jüttler et al., 2022). Internal agents may gain access to the data preprocessing (*blackbox access*) or the training pipeline itself (*whitebox access*) (Lee, 2022; Wang et al., 2015). Furthermore, in sensitive domains like healthcare and finance, modifying data values is often monitored through integrity checks and anomaly detection (Vimalachandran et al., 2016; on Banking Supervision, 2013; Mashrur et al., 2020). In such settings, data ordering attacks pose a stealthy, alternative threat to traditional poisoning methods.

The success of data ordering attacks depends on the alignment between clean gradients and adversarial gradients (Souri et al., 2022; Lederer et al., 2023). Attack algorithms reorder clean samples so that their gradient updates mimic those induced by adversarial samples. However, prior work using

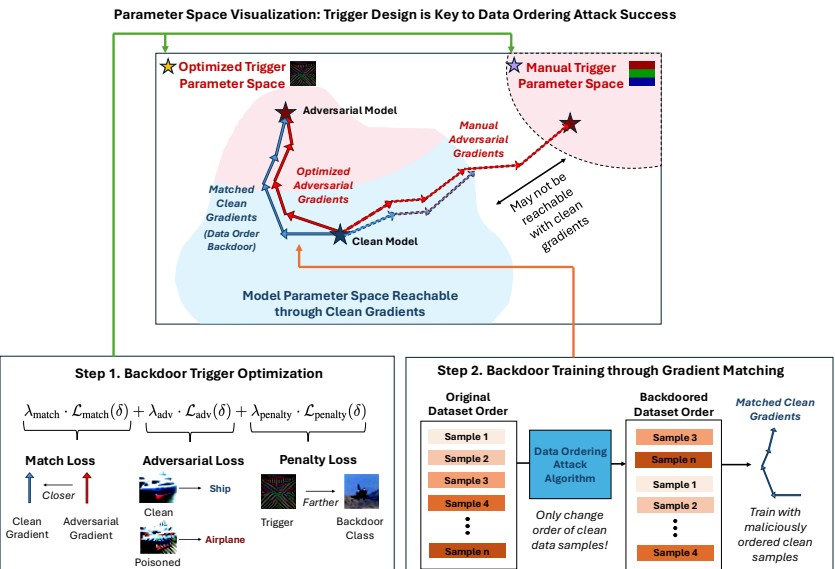

Figure 1: Overview of our TOGA pipeline. Within a single training epoch, only a subset of parameter space (blue) is reachable through clean gradients. (Step 1) TOGA optimize a trigger such that the backdoor-effective space (red) lie within the reachable space (blue). Trigger optimization uses three losses: match loss aligns adversarial and clean gradients, adversarial loss encourages prediction of backdoor class on poisoned inputs, and penalty loss prevents collapse to a trivial target-class sample. (Step 2) Clean samples are then reordered to match the adversarial gradients, creating a poisoned training sequence without modifying inputs or labels.

manually chosen triggers can fail to achieve sufficient gradient alignment, limiting attack success (Shumailov et al., 2021). The parameter regions effective for manual triggers may lie outside the space reachable by clean gradient trajectories (Figure 1).

As a step forward, we present **Trigger Optimization for Gradient Alignment (TOGA)**, which improves attack success by explicitly optimizing triggers (Step 1) that maximize gradient alignment (Step 2, Figure 1). This optimization increases the practical danger of clean data ordering attacks by significantly improving their effectiveness and specificity. We evaluate TOGA on three standard benchmarks — CIFAR-10 (Krizhevsky et al., 2009), CelebA (Liu et al., 2015), and ImageNet (Deng et al., 2009) — and two sensitive domain datasets: ISIC 2018 for melanoma classification (Codella et al., 2019) and UCI Credit-g for credit prediction (Hofmann, 1994). Our optimized triggers outperform existing baselines, improving the average whitebox attack success rate (ASR) by 46% and achieving up to 99% ASR on ISIC and 92% on CelebA, all with $< 5\%$ drop in benign accuracy.

Additionally, we investigate subpopulation-specific backdoors. Prior work has shown that vulnerable subgroups can be disproportionately impacted by targeted poisoning (Jagielski et al., 2021; Lin et al., 2020; Kulkarni et al., 2024). We show that TOGA can be extended to this setting, learning triggers that disproportionately affect subpopulations. TOGA reaches 57% ASR in credit prediction when targeting socioeconomic attributes like employment, and 99% ASR in melanoma classification by exploiting artifacts such as ink markings. To our knowledge, this is the first demonstration of subpopulation backdoor attacks via data ordering.

Our key contributions are:

1. We introduce a novel trigger optimization framework (TOGA) that learns effective triggers for data ordering attacks and significantly outperforms existing trigger-selection baselines across five benchmarks.

2. We show that TOGA triggers can selectively target subpopulations. For example, TOGA backdoors can assign poor credit scores to unemployed individuals or cause underdiagnosis of melanoma in patients with physical artifacts like hair.

3. We show that TOGA remains effective under standard purification and gradient-similarity defenses, which either fail to significantly reduce ASR or do so only at the cost of severe drops in benign accuracy.

## 2 RELATED WORK

### 2.1 BACKDOOR ATTACKS

Backdoor attacks are a type of adversarial attack in which a model is trained to produce a malicious output whenever a specific trigger pattern appears in the input (Gu et al., 2017; Li et al., 2022). BadNets (Gu et al., 2017) introduced backdoor attacks in neural networks by poisoning data with a visible trigger paired with a backdoor label. Since then, researchers have developed increasingly stealthy and robust backdoor attacks that remain effective under diverse conditions, such as physical-world constraints (Li et al., 2021a; Xue et al., 2021; Gong et al., 2023) and model transfer settings (Zhang et al., 2022; Feng et al., 2022). One direction of research focuses on invisible attacks, where imperceptible triggers still reliably activate the backdoor (Chen et al., 2017; Zhong et al., 2020; Li et al., 2021b; Gao et al., 2024). Another category involves data poisoning without label modification, known as "clean-label" attacks (Turner et al., 2019; Geiping et al., 2020; Souri et al., 2022). Overall, these studies assume adversaries can modify the training data, a strategy likely to be detected in sensitive domains with standard integrity checks (Kissi et al., 2023; Vimalachandran et al., 2016; Katari & Ankam, 2022; Hilal et al., 2022; Ahmed et al., 2016). To address this limitation, we investigate a less perceptible approach known as data ordering attacks.

### 2.2 DATA ORDERING BACKDOOR ATTACKS

Shumailov et al. (2021) first introduced data ordering attacks via gradient alignment, where backdoor behavior is induced by matching clean batch gradients with poisoned gradients. Ganesh et al. (2023) showed that even a single epoch of data reordering can negatively affect subpopulation fairness. Liu et al. (2024) examined how data ordering affects generalization error in federated learning. Crucially, all prior data ordering backdoor methods use *a fixed, manually-chosen trigger*. Our approach improves the attack success rate by *learning an optimal trigger* for the given training data.

### 2.3 OPTIMIZED BACKDOOR ATTACKS

Of relevance to our work are approaches that explicitly optimize backdoor triggers to enhance attack performance (Saha et al., 2020; Tao et al., 2022; Arshad et al., 2024). Li et al. (2020) optimizes triggers by scaling neuron activations toward target values. Doan et al. (2021b) minimizes a balance of clean and adversarial training losses. Optimized backdoors have also been applied in the settings of federated learning (Zhang et al., 2023; Yang et al., 2023), contrastive learning (Sun et al., 2024; Liang et al., 2024), and latent space manipulation (Doan et al., 2021a; Zhao et al., 2022). However, to our knowledge, no existing method has demonstrated the effectiveness of trigger optimization for data ordering attacks.

### 2.4 SUBPOPULATION-BASED SEMANTIC BACKDOOR ATTACKS

Another class of backdoor attacks uses semantic features as triggers to target specific subpopulations (Li et al., 2021b; Khaddaj et al., 2023). For instance, natural physical traits can act as unintentional triggers (Wenger et al., 2022). Composite attacks combine multiple benign features to activate backdoors in both vision (Lin et al., 2020) and natural language tasks (Huang et al., 2023). Subpopulation poisoning attacks similarly exploit feature clustering to target subgroups with similar characteristics (Jagielski et al., 2021). These approaches assume a different setting, where the adversary can poison training data but lacks control at inference. In contrast, our setting assumes clean training data, with the adversary acting only at inference time.

## 3 METHOD

### 3.1 THREAT MODEL

We first define the actors in our threat model. The victim is a benign party seeking to train a clean model (Joe et al., 2022; Hanif et al., 2024). The attacker is a malicious agent who reorders training samples but does not modify their values (Shumailov et al., 2021). We consider two attacker variants. The first is a blackbox attacker, who lacks knowledge of the model architecture and training configuration (Bai et al., 2023). In this blackbox setting, we use surrogate models with fewer parameters (e.g. ResNet-18 as surrogate for VGG-16). The second is a whitebox attacker with access to the model under attack, allowing gradient computation (Gil et al., 2019).

Our TOGA threat model builds on Shumailov et al. (2021), but with more relaxed assumptions for attacker access. While Shumailov et al. (2021) assumes full control over the training order and introduces adversarial batches throughout training, TOGA only requires access to: (1) one clean training epoch to collect the dataset for trigger optimization, and (2) control over sample order during the final training epoch. These conditions allow TOGA to induce a backdoor using adversarially ordered *clean samples* during the *last epoch* only. With only one adversarial epoch needed, TOGA is also functional in an offline setting. If the attacker can infer the random seed for the data generator, they can precompute the adversarial order and overwrite the static data file to enforce that order in the final epoch.

### 3.2 PROBLEM FORMULATION

We define a model $F(x, \theta) = y$ with input $x \in \mathbb{R}^n$, output $y \in \mathbb{R}^m$, and parameters $\theta \in \mathbb{R}^p$ with loss function $\mathcal{L}$. We define the clean training dataset as $X = \{(x_i, y_i)\}_{i=1}^N$. To compute the objective functions for trigger optimization and gradient alignment, we introduce a set of adversarial samples $X^{adv} = \{(x_i^{adv}, y_i^{adv})\}_{i=1}^N$. Importantly, these adversarial samples are *not* used in any training for backdoors, but solely as reference points to guide optimization for data ordering. As with prior work (Shumailov et al., 2021; Souri et al., 2022), we assume an all-to-one backdoor (Li et al., 2022) with universal trigger $\delta$. Specifically, there is a single target class $y_i^{adv} = y^{adv}$ and the adversarial samples take the form of $x_i^{adv} = x_i + \alpha\delta$, with $\alpha$ representing the trigger strength.

We wish to optimize the trigger $\delta$ and model $\theta$ to minimize both clean and adversarial losses. Formally, we define the bi-level optimization problem:

$$\min_{\delta \in D} \sum_{i=1}^N \mathcal{L}(F(x_i + \alpha\delta, \theta(\delta)), y^{adv}) \quad \text{s.t.} \quad \theta(\delta) \in \arg\min_\theta \frac{1}{N} \sum_{i=1}^N \mathcal{L}(F(x_i, \theta(\delta)), y_i)$$

### 3.3 OUR APPROACH

Our approach consists of two sequential components: 1) optimizing a trigger pattern that facilitates data ordering attacks (Figure 1, Step 1), and 2) determining an ordering of the training dataset such that clean sample gradients align with adversarial gradients, allowing effective backdoor learning (Figure 1, Step 2). Part 1 introduces a novel framework for trigger optimization. Part 2 adapts the formulation from Shumailov et al. (2021), and proposes a greedy algorithm to solve the gradient alignment problem.

**Trigger Optimization through Regularized Losses** We begin by training a model $F(x, \theta)$ on clean data. In the whitebox case, we have access to the victim model, allowing us to compute gradients and losses with respect to it. In the blackbox case, we train a lower-capacity surrogate model until its validation accuracy converges, and use it to guide the optimization. We fix the clean model parameters $\theta^* = \arg\min_\theta \frac{1}{N} \sum_{i=1}^N \mathcal{L}(F(x_i, \theta), y_i)$. Now, we optimize the universal perturbation trigger $\delta$ using the following regularized objective:

$$\min_{\delta \in \mathbb{R}^d} \quad \lambda_{\text{match}} \cdot \mathcal{L}_{\text{match}}(\delta) + \lambda_{\text{adv}} \cdot \mathcal{L}_{\text{adv}}(\delta) + \lambda_{\text{penalty}} \cdot \mathcal{L}_{\text{penalty}}(\delta)$$
$$\text{subject to} \quad \|\delta\|_\infty \leq \epsilon \tag{1}$$

where the three loss terms are defined as:

$$\mathcal{L}_{\text{match}}(\delta) = \frac{1}{D} \left\| \nabla_\theta \mathcal{L}\left(F(x_i, \theta^*), y_i\right) - \nabla_\theta \mathcal{L}\left(F(x_i + \alpha\delta, \theta^*), y^{\text{adv}}\right) \right\|_2^2$$

$$\mathcal{L}_{\text{adv}}(\delta) = \mathcal{L}\left(F(x_i + \alpha\delta, \theta^*), y^{\text{adv}}\right)$$

$$\mathcal{L}_{\text{penalty}}(\delta) = 1 + \frac{\delta \cdot \mu}{\|\delta\| \, \|\mu\|}, \quad \mu = \frac{1}{K}\sum_{i=1}^{K} x_i, \quad x_i \sim \mathcal{D}(x|y = y^{\text{adv}})$$

The proposed optimization approximates the bi-level problem by fixing model parameters $\theta^*$ and thus decoupling trigger optimization from model training. Given a trained clean model, the success of a data ordering attack depends on aligning clean and adversarial gradients to steer the model toward backdoor learning. The first term, $\mathcal{L}_{\text{match}}$, explicitly encourages this alignment by minimizing the mean squared distance between individual clean and adversarial gradients (see Appendix D for derivation). The second term, $\mathcal{L}_{\text{adv}}$ (adversarial loss), directly optimizes attack success by maximizing the model's misclassification rate on adversarial inputs. The third term, $\mathcal{L}_{\text{penalty}}$, regularizes the trigger to prevent collapse to the prototypical backdoor class (e.g., an airplane for the airplane class). The loss penalizes cosine similarity between the trigger $\delta$ and the mean feature vector $\mu$ of randomly sampled target class images. Additionally, to prevent the trigger from overwhelming the input and collapsing all poisoned samples to a single prototype, we enforce an $\ell_\infty$ norm constraint that bounds the maximum perturbation magnitude.

**Trigger Optimization with Subpopulations** Now, we extend our trigger optimization formulation to a subpopulation-aware setting, where the goal is to optimize a trigger that is only effective within a specific subpopulation. Specifically, we want the model to predict the adversarial target class when it sees both the trigger $\delta$ and a semantic feature $\phi$ associated with a predefined subpopulation (e.g., "single male" or "unemployed" in credit prediction). Outside of the subpopulation, the trigger should not induce misclassification.

To achieve this, we introduce two modifications: (1) a subpopulation-aware separability loss $\mathcal{L}_{\text{subpop}}$ that encourages the model to predict the target class within the subpopulation, and (2) a penalty term $\mathcal{L}_{\text{spillover}}$ to discourage the model from predicting the target class outside the subpopulation.

The full optimization objective becomes:

$$\min_{\delta \in \mathbb{R}^d} \quad \lambda_{\text{match}} \cdot \mathcal{L}_{\text{match}}(\delta) + \lambda_{\text{subpop}} \cdot \mathcal{L}_{\text{subpop}}(\delta)$$
$$+ \lambda_{\text{spillover}} \cdot \mathcal{L}_{\text{spillover}}(\delta) + \lambda_{\text{penalty}} \cdot \mathcal{L}_{\text{penalty}}(\delta) \tag{2}$$
$$\text{subject to} \quad \|\delta\|_\infty \leq \epsilon$$

The match loss $\mathcal{L}_{\text{match}}$ is the same. We modify the attack success objective to condition on subpopulation membership. The subpopulation-aware attack loss is defined as:

$$\mathcal{L}_{\text{subpop}}(\delta) = \mathcal{L}\left(F(x_i + \alpha\delta, \theta^*), \, z_i \cdot y^{\text{adv}}\right)$$

where $z \in \{0, 1\}$ is the binary subpopulation membership indicator (with $z = 1$ if the sample belongs to the targeted subpopulation). This loss encourages the model to predict the adversarial class only within the intended subpopulation.

To discourage the model from spuriously predicting the target class outside the subpopulation, we define an explicit spillover penalty:

$$\mathcal{L}_{\text{spillover}}(\delta) = \mathbb{I}[z = 0 \wedge y_i \neq y^{\text{adv}}] \cdot F(x + \alpha\delta, \theta^*)_{y^{\text{adv}}}$$

This term penalizes the model's confidence in the adversarial target class ($F(x + \delta, \theta^*)_{y^{\text{adv}}}$) on poisoned inputs that do not belong to the subpopulation ($z = 0$) and whose original label differs from the adversarial class ($y_i \neq y^{\text{adv}}$). Finally, the regularization term $\mathcal{L}_{\text{penalty}}$ remains unchanged.

To summarize, this formulation allows us to learn subpopulation-specific backdoor triggers that are stealthier and more targeted, activating only when both the trigger and the semantic attribute of the targeted subpopulation are present.

**Backdoor Training through Gradient Alignment** To train the model to learn the backdoor, we use a similar gradient alignment formulation to Shumailov et al. (2021). The objective is to reorder data such that clean batch gradients closely approximate the adversarial batch gradients:

$$\nabla L_{batch}(F(x_i, \theta(\delta)), y_i) \approx \nabla L_{batch}(F(x_i + \alpha\delta, \theta(\delta)), y^{adv})$$
$$\nabla L(X_i) \approx \nabla L(X_j^{adv})$$

This gradient similarity can be formalized as a norm minimization problem. Given a set of poisoned batches $X_j^{adv}$, we aim to find the corresponding clean batches $X_i$ that minimizes the distance between their gradient vectors:

$$\min_{X_i} ||\nabla L(X_i) - \nabla L(X_j^{adv})||^p \tag{3}$$

We solve this optimization problem through a greedy heuristic, with the exact algorithm detailed in Appendix E. To summarize, the algorithm uses the $L_p$ norm between individual clean samples $x_i$ and adversarial gradient $X_j^{adv}$ to greedily assign each $x_i$ to construct the clean batches.

## 4 TOY EXAMPLE: TRIGGER DESIGN IS KEY TO DATA ORDERING ATTACK SUCCESS

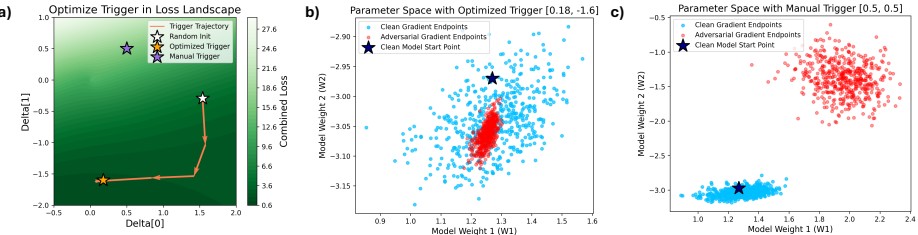

Figure 2: Toy example of a linear classifier on the 2D make_moons dataset. (a) Loss landscape of the trigger optimization. Orange arrows trace the learned trigger from random initialization (white star) to optimized TOGA trigger (orange star); manual trigger at [0.5, 0.5] (purple star). (b-c) Parameter spaces for the (b) TOGA and (c) manual triggers corresponding to Figure 1. We simulate 500 clean and adversarial gradient trajectories over one epoch and plot the final model weights: blue for clean and red for adversarial paths.

As a toy example, we illustrate the importance of trigger design with a linear classifier on the make_moons dataset (Pedregosa et al., 2011). This setting is intentionally chosen so that the input, trigger, model parameters, and gradients are all two-dimensional, enabling direct visualization. The trained clean model has weights [1.27, -2.97]. We optimize a trigger using this fixed clean model and obtain [0.18, -1.60] (Figure 2a). For comparison, we also evaluate a manually selected trigger [0.5, 0.5]. In Figure 2b, we observe that adversarial training with the TOGA trigger leads to backdoored model weights (red dots) that overlap with clean gradient model weights (blue dots). The manual trigger, in contrast, drives model trajectories toward regions not attainable by clean gradients (red dots separate from blue dots in Figure 2c). These results illustrate that without careful trigger selection, it may be impossible to reach adversarial model parameters with high backdoor performance using only clean model gradients.

## 5 EXPERIMENTAL SETUP

### 5.1 DATASETS

We evaluate our trigger optimization framework across five datasets: three natural image datasets (CIFAR-10 (Krizhevsky et al., 2009), CelebA (Liu et al., 2015), and ImageNet (Deng et al., 2009))

and two sensitive-domain datasets (ISIC 2018 (Codella et al., 2019) and UCI Credit-g (Hofmann, 1994)). For each dataset, we define a specific backdoor for each benign classification task:

**CIFAR-10**: The backdoor task is to predict the "airplane" class (class 0), while the benign task is standard 10-way image classification.
**CelebA**: The backdoor task is to predict "smiling" (class 1), with the benign task being binary classification of whether the person is smiling.
**ImageNet**: The backdoor task is to predict class 954 ("banana"), with the benign task being the ILSVRC-2012 1000-way classification.
**ISIC 2018**: The backdoor task is to predict non-malignant (class 0), while the benign task is melanoma classification. We select non-malignancy as the target class because misclassifying melanoma as non-malignant (i.e., underdiagnosis) poses a greater clinical risk (Cockburn et al., 2008; Cassalia et al., 2024).
**Credit-g**: The backdoor task is to predict bad credit (class 0), while the benign task is binary classification of good versus bad credit risk given a person's financial profile.

### 5.2 MODELS

To compare with prior work (Shumailov et al., 2021), we adopted VGG-16 (Simonyan & Zisserman, 2014) as the base (whitebox) model and ResNet-18 (He et al., 2016) as the surrogate (blackbox) model for CelebA. We also conducted larger scale experiments with ViT-16 models (Dosovitskiy et al., 2020) on ImageNet. For ISIC 2018, we use DenseNet121 (Huang et al., 2017) as the surrogate model and ResNet-50 (He et al., 2016) as the base model, both of which are standard architectures in dermatology imaging (Anand et al., 2022; Yadav et al., 2024). See Appendix A for details.

To evaluate backdoor performance, we use benign accuracy and attack success rate. Benign accuracy (Li et al., 2021a;b) is the classification accuracy on clean, unmodified test inputs, measuring the model's performance on its original task. Attack success rate (ASR) (Li et al., 2021a;b) is the proportion of adversarial inputs that are misclassified into the attacker-specified target class, reflecting the effectiveness of the backdoor trigger. For subpopulation backdoors, we define an additional metric: outgroup accuracy. It measures the accuracy of predicting the original label for poisoned inputs *outside* the subpopulation, reflecting backdoor specificity. See Appendix B for details.

## 6 RESULTS

### 6.1 OPTIMIZED TRIGGER PATTERNS IMPROVE BACKDOOR PERFORMANCE

We evaluate on three natural image datasets (CIFAR-10, CelebA, and ImageNet) and two sensitive domain datasets (ISIC 2018 and Credit-g). Our method ("TOGA") is compared against two baselines (Figure 4): a flag trigger used in prior work ("Flag") (Shumailov et al., 2021) and a representative sample from the target backdoor class ("Class"). For the tabular Credit-g dataset (61 features), we adapt the visual "flag" trigger into a banded perturbation: apply -0.3 to the first 9 columns, +0.3 to the last 9, and 0 elsewhere, yielding a flag-like pattern. We choose 9 columns on each side so the total perturbation magnitude matches that of our TOGA trigger.

We observe **a substantial improvement in attack success rate (ASR) with our optimized TOGA trigger compared to baseline triggers across the all five datasets** (Table 1). For image datasets, the improvement is the most pronounced on CIFAR10, where our method achieves a whitebox ASR of 99.39%, substantially outperforming both the flag trigger (25.51%) and class trigger (17.27%). In the tabular Credit-g dataset, our method also achieves strong results, with a whitebox ASR of 87.5% and blackbox ASR of 80.2%, compared to much lower ASRs from the class trigger (22.9% whitebox, 25.9% blackbox) and the flag trigger (13.3% whitebox, 17.4% blackbox). In all TOGA attacks, whitebox ASR outperforms blackbox ASR, as expected due to the surrogate model's limited ability to replicate the true model. Our method also maintains benign accuracy comparable to the baselines, with minor accuracy drops ($< 5\%$) relative to the clean model.

Besides manual triggers, we compare TOGA against a competitive baseline from Silent Killers (Lederer et al., 2023). We reproduced their trigger optimization results on CIFAR-10 with their provided ResNet-18 model. We observe an ASR of 91.1% (vs. their reported 90.65%). Under the same setup, our TOGA framework achieves 98.3% ASR, demonstrating stronger attack performance.

Together, these results demonstrate that trigger optimization significantly enhances backdoor effectiveness without compromising clean model performance. In sensitive domains, TOGA allows attackers to induce harmful behaviors, such as denying credit to qualified individuals or underdiagnosing melanoma, without manual tuning or domain expertise. These findings highlight the practical threat of trigger-optimized data ordering backdoors.

| Dataset | Trigger | Setting | Benign Acc (↑) | Change in Benign Acc (↑) | ASR (↑) |
|---------|---------|---------|----------------|--------------------------|---------|
| CIFAR10 | Flag | WhiteBox | 93.31 ± 0.17 | -0.45 ± 0.17 | 25.51 ± 1.57 |
| | | BlackBox | 93.77 ± 0.38 | 0.01 ± 0.38 | 11.42 ± 2.98 |
| | Class | WhiteBox | 93.40 ± 0.08 | -0.36 ± 0.08 | 17.27 ± 0.80 |
| | | BlackBox | 93.78 ± 0.21 | -0.05 ± 0.15 | 10.45 ± 1.81 |
| | TOGA | WhiteBox | 93.67 ± 0.39 | -0.09 ± 0.39 | **99.39 ± 0.22** |
| | | BlackBox | 93.21 ± 0.34 | -0.56 ± 0.34 | **79.05 ± 1.69** |
| CelebA | Flag | WhiteBox | 92.64 ± 0.16 | -0.02 ± 0.21 | 44.93 ± 1.41 |
| | | BlackBox | 92.08 ± 0.26 | -0.58 ± 0.30 | 46.13 ± 1.20 |
| | Class | WhiteBox | 92.65 ± 0.15 | -0.01 ± 0.21 | 49.03 ± 1.21 |
| | | BlackBox | 92.08 ± 0.23 | -0.58 ± 0.27 | 51.40 ± 1.03 |
| | TOGA | WhiteBox | 91.94 ± 0.15 | -0.72 ± 0.21 | **92.07 ± 1.43** |
| | | BlackBox | 92.05 ± 0.30 | -0.61 ± 0.33 | **79.61 ± 2.74** |
| ImageNet | Flag | WhiteBox | 81.06 ± 0.01 | -0.01 ± 0.01 | 38.89 ± 0.62 |
| | | BlackBox | 80.88 ± 0.01 | -0.18 ± 0.01 | 10.99 ± 0.47 |
| | Class | WhiteBox | 81.05 ± 0.01 | -0.01 ± 0.01 | 47.12 ± 0.87 |
| | | BlackBox | 80.90 ± 0.01 | -0.16 ± 0.01 | 31.66 ± 0.72 |
| | TOGA | WhiteBox | 80.94 ± 0.04 | -0.12 ± 0.03 | **89.15 ± 0.47** |
| | | BlackBox | 80.91 ± 0.00 | -0.16 ± 0.00 | **72.90 ± 0.65** |
| ISIC (Derm) | Flag | WhiteBox | 83.03 ± 0.89 | 1.88 ± 1.72 | 87.70 ± 3.04 |
| | | BlackBox | 83.46 ± 1.24 | 2.31 ± 1.92 | 84.92 ± 10.56 |
| | Class | WhiteBox | 82.92 ± 1.11 | 1.77 ± 1.84 | 91.48 ± 5.51 |
| | | BlackBox | 83.23 ± 1.13 | 2.08 ± 1.85 | 86.20 ± 9.98 |
| | TOGA | WhiteBox | 83.00 ± 1.19 | 1.85 ± 1.89 | **99.46 ± 0.74** |
| | | BlackBox | 83.42 ± 1.11 | 2.27 ± 1.84 | **99.07 ± 1.08** |
| Credit-g | Flag | WhiteBox | 94.10 ± 1.13 | -1.30 ± 1.19 | 13.30 ± 3.63 |
| | | BlackBox | 95.00 ± 0.43 | -0.40 ± 0.56 | 17.40 ± 3.22 |
| | Class | WhiteBox | 93.60 ± 0.72 | -1.80 ± 0.80 | 22.90 ± 0.78 |
| | | BlackBox | 95.00 ± 0.43 | -0.40 ± 0.56 | 25.90 ± 2.06 |
| | TOGA | WhiteBox | 94.80 ± 0.24 | -0.60 ± 0.43 | **87.50 ± 5.15** |
| | | BlackBox | 94.80 ± 0.49 | -0.60 ± 0.61 | **80.20 ± 5.61** |

Table 1: Benign accuracy and ASR for different triggers. The "Change in Benign Acc" column shows the difference in benign accuracy before and after backdoor training (negative values indicate a drop in accuracy). $2\sigma$ CIs computed over 5 seeds.

## 6.2 Backdoor Triggers Are Optimizable to Target Vulnerable Subpopulations

We extend our trigger optimization framework to support subpopulation-specific backdoors, optimizing triggers that selectively target vulnerable or sensitive subgroups. To reflect more realistic attack scenarios, we focus on blackbox settings and apply our method to the sensitive domain datasets. For the Credit-g dataset, we define subpopulations based on socioeconomic features such as employment (e.g. unemployed), housing (e.g. rent), and personal status (e.g. female married). For the ISIC dermatology dataset, we target physical artifacts (e.g. ink, ruler, hair) that can be intentionally introduced into dermoscopic images.

Our TOGA triggers achieve consistently higher ASR than baseline patterns (Figure 3). TOGA triggers can achieve up to 63.2% ASR in Credit-g and 99.4% ASR for ISIC dataset. Outgroup accuracy remains comparable in ISIC and slightly lower in Credit-g, indicating strong backdoor specificity. Benign accuracy is stable across all triggers.

These results demonstrate that **our TOGA trigger improves attack success while maintaining subpopulation specificity and benign performance.** Crucially, this work is the first to show that

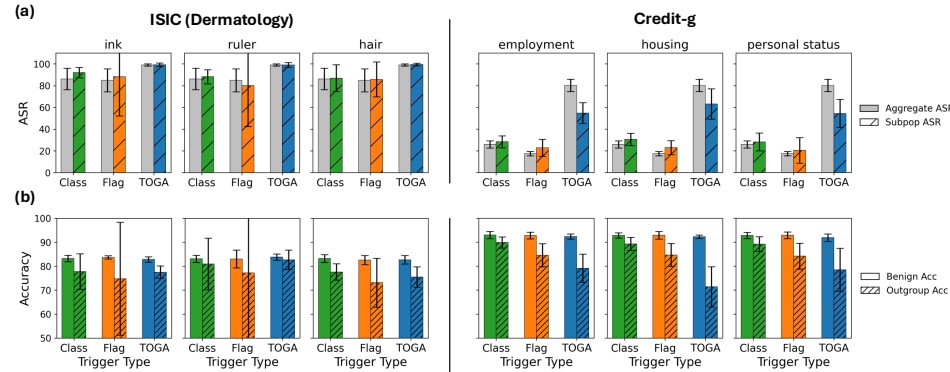

Figure 3: Barplot of (a) ASR, and (b) benign and outgroup accuracy of subpopulation backdoors. Aggregate ASRs from Table 1 are provided for comparison. Error bars show $2\sigma$ CIs over 5 seeds.

data ordering attacks can be adapted to target vulnerable subpopulations in high-stakes domains, raising new risks for model integrity and fairness.

### 6.3 DEFENSES AND ROBUSTNESS

We evaluated TOGA against three defense strategies: inference-time JPEG compression (Das et al., 2018), a diffusion-based data purification method (PureGen) (Pooladzandi et al., 2024), and a gradient similarity-based detection method (Dhaliwal & Shintre, 2018).

**Attack Purification** JPEG compression is largely ineffective as a defense (Table 4). On CelebA, ASR remains high (71%) even at an extreme quality of 20%. On ImageNet, TOGA remains effective until similarly aggressive compression, but at this point benign accuracy collapses from 81% to 71%, making it impractical as an inference-time defense. PureGen, a state-of-the-art purification method combining EBMs and diffusion models, was adapted to training time so that clean batches (though still adversarially ordered) were purified. TOGA remains robust (>90% ASR) under Pure-Gen until extreme purification of 10,000 EBM steps, which reduces ASR to 65% but also drops benign accuracy on CIFAR10 from 93% to 73% (Table 5). In summary, neither defense reduces ASR without severely degrading benign accuracy.

**Attack Detection** Because TOGA selects more "adversarial" batches of clean data, we evaluate a gradient-similarity defense that classifies gradients using their norm and maximum cosine similarity. Following prior work, the Unknown-Trigger setting substitutes a proxy attack (FGSM), but detection accuracy against TOGA is low (30% and 42%), allowing the attacker to succeed by increasing matched adversarial batches (Table 6). Detection improves in the Known-Trigger setting (72% and 70%), but this assumes defender access to the true trigger. In realistic scenarios, TOGA remains effective despite such defenses.

Lastly, we evaluate ablations of each loss term in our trigger optimization (Supplementary Section G). Removing the gradient matching loss ($\lambda_{\text{match}}$) reduces ASR from 87.5% to 74.6%, while removing the adversarial loss ($\lambda_{\text{adv}}$) causes a sharp drop to 36.8%. Excluding the penalty term ($\lambda_{\text{penalty}}$) increases the cosine similarity between the trigger and the target class prototype from -0.317 to 0.227, indicating a tendency to overfit to trivial patterns.

## 7 CONCLUSION

Data ordering attacks pose an emerging threat by enabling backdoors through clean data manipulation alone. We show that optimizing the trigger pattern substantially increases attack success rates and enables subpopulation-specific targeting. These findings highlight the need for order-aware defenses and fairness evaluations that account for subpopulation vulnerabilities, particularly in high-stakes domains.

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
