# OpenReview forum: "TOGA: Trigger Optimization for Clean Data Ordering Backdoor Attack"
_ICLR.cc/2026/Conference — Submitted to ICLR 2026_

### Official Review · Reviewer_6LNn · 2025-10-21

**Soundness:** 3
**Presentation:** 2
**Contribution:** 2
**Rating:** 2
**Confidence:** 4

**Summary:**

The paper proposes an approach for optimizing backdoor poisoning by data-ordering attack.
The attack aims to perturb the gradient of a data-batch during deep learning (back-propagation).

**Strengths:**

The authors propose an interesting optimization framework with several penalty terms indicating that they understand all of the issues in play.

**Weaknesses:**

First note that data-ordering attacks require the adversary to be an insider of the training process to affect the batch gradients used for deep learning and to have access to the complete training dataset (Eq. (3)). Though the authors argue that their attack requires less assumptions than previously proposed data-ordering attacks (see line 173-), their attack nevertheless requires much stronger assumptions than passive data-poisoning (esp. prior to the formation of the training dataset) as considered in most backdoor papers. Moreover, a benign training authority (if present) could simply randomly re-select from the training data to form batches to defeat any data-ordering attack. So, it's not clear to me how practically important such attacks are. Though it's interesting how data ordering could bias a model, prior papers have already established this.

The training of models described in 208 and optimizations such as (3) raise additional concerns regarding the adversary's work-factor relative to an obvious and simple defense (or to simple passive data-poisoning attack).

On line 191, I think the authors should point out that the ground truth class of x_i is _not_ y^{adv}, which is confused by writing y_i^{adv}=y^{adv}.

The description of the method on p. 5-6 is pretty terse.

Lines 244, 259 and 260 say the same thing.
The fonts on the figures are too small.
:

**Questions:**

See above.

---

> ### Author Response · Authors · 2025-12-04
> **Clarification and Corrections**
>
> We thank the reviewer for their thoughtful feedback and constructive comments.
>
> **Novelty and practicality of data-ordering attacks.**
> We agree that data-ordering attacks assume a different threat model from standard data poisoning. Our focus is specifically on insider adversaries, which is increasingly relevant in modern large-scale training pipelines where training is distributed or handled by multiple teams. Within this setting, data-ordering attacks are uniquely concerning because (1) they require no poisoned inputs, (2) they modify only the ordering of clean data, and (3) they operate using a single adversarial epoch, making them difficult to detect. Although dynamic batch re-sampling is a possible defense, it is not standard in most training pipelines, which typically rely on dataloaders with fixed seeds. We view our work as complementary to existing poisoning research by highlighting a distinct and understudied insider threat.
>
> **Line 191 (notation correction).**
> We will clarify that $y_{\text{adv}}$ denotes the chosen adversarial class, and that the clean labels $y_i$ are not equal to $y_{\text{adv}}$.
>
> **Method description (p. 5–6).**
> We will expand the method section to provide clearer intuition and a more explicit explanation of both the trigger-optimization objective and the gradient-alignment procedure.
>
> **Repeated statements (lines 244, 259, 260).**
> We will remove redundant explanations of the subpopulation penalty term.
>
> **Figure readability.**
> We will increase font sizes and improve figure clarity in the revised manuscript.

---

### Official Review · Reviewer_KZ8X · 2025-10-26

**Soundness:** 2
**Presentation:** 2
**Contribution:** 2
**Rating:** 4
**Confidence:** 4

**Summary:**

This paper introduces TOGA, a novel framework for optimizing backdoor triggers specifically tailored to clean data ordering attacks. Rather than modifying training data directly, TOGA identifies triggers whose adversarial gradient trajectories are well-aligned with those reachable by merely reordering clean training samples. The approach is empirically evaluated across diverse datasets (CIFAR-10, CelebA, ImageNet, ISIC 2018, UCI Credit-g), demonstrating higher attack success rates than prior baselines while maintaining clean performance. The framework is also extended to subpopulation-specific backdoors and tested against recent defenses.

**Strengths:**

+ Advancement over Prior Work. TOGA addresses a missing piece in data ordering attacks, by proposing a concrete, optimization-based trigger-selection framework. This idea is substantive, as previous data ordering techniques largely assumed manually chosen triggers, limiting attack effectiveness
+ Extensive Evaluation. The experimental results are comprehensive, spanning image and tabular domains, and including sensitive applications.
+ Extension to Subpopulation Attacks. The method is extended to design subpopulation-targeted triggers, providing a new threat axis toward vulnerable demographic groups.
+ Reproducibility. Supplied materials code is attached for reproducibility.

**Weaknesses:**

A substantive assessment of the weaknesses of the paper. Focus on constructive and actionable insights on how the work could improve towards its stated goals. Be specific, avoid generic remarks. For example, if you believe the contribution lacks novelty, provide references and an explanation as evidence; if you believe experiments are insufficient, explain why and exactly what is missing, etc.

+ Incremental Advancement Over Data Ordering Pipelines. While TOGA provides a principled mechanism for trigger optimization within data ordering attacks, the core pipeline remains largely based on Shumailov et al.’s gradient alignment framework [1]. The paper would benefit from a clearer articulation of conceptual novelty beyond trigger search, and ablations that isolate how much of the reported gains stem specifically from the proposed optimization rather than broader design choices or hyperparameter tuning.
+ Ambiguity and Potential Flaws in Mathematical Formulation. In Section 3.3, several aspects are under-specified or ambiguous:
	+ In the definition of $\mathcal{L}_{\text{match}}$, the variable $D$ appears in the normalization term but is not explicitly defined in the local context (e.g., batch size vs. dataset cardinality). Clarifying this would improve reproducibility and reader understanding.
	+ In the subpopulation-aware objective $\mathcal{L}_{\text{subpop}}$, the use of $z_i \cdot y^{\mathrm{adv}}$ as a target encoding is somewhat unconventional. A concise explanation of how this interacts with standard label representations (e.g., masking vs. mixing) would prevent potential misinterpretation.
	+ For the spillover penalty $\mathcal{L}_{\text{spillover}}$, penalizing the target logit directly rather than the full loss may not necessarily correlate with reduced misclassification risk. An ablation comparing loss-based versus logit-based regularization would strengthen confidence in this design choice.
+ Limited Analysis of Clean-Benign Tradeoffs. Although the paper reports a “<5%” drop in clean accuracy, a more granular analysis would be helpful. In particular, it is unclear whether certain fine-grained classes, minority attributes, or distributional subgroups are disproportionately affected. Rare catastrophic degradations could matter significantly in security-sensitive deployments.
+ Assumptions on Adversarial Capabilities. The threat model assumes an adversary with substantial visibility into training loss trajectories. While plausible in compromised supply-chain scenarios, a more explicit discussion of when such access is feasible would contextualize applicability. Additionally, comparison to alternative stealth vectors (e.g., weight manipulation under constrained visibility) would strengthen the realism argument.
+ Evaluation Model Specification and Training Details. The manuscript does not clearly specify the exact architectures and training hyperparameters used for each dataset.


[1] Ilia Shumailov, Zakhar Shumaylov, Dmitry Kazhdan, Yiren Zhao, Nicolas Papernot, Murat A. Erdogdu, and Ross Anderson. 2021. Manipulating SGD with data ordering attacks. In Proceedings of the 35th International Conference on Neural Information Processing Systems (NIPS '21)

**Questions:**

1.Ablation on Optimization Components

Could the authors provide ablations isolating contributions from (i) trigger optimization, (ii) spillover penalty, and (iii) subpopulation-aware loss? Without this, it is difficult to attribute gains to the proposed components.

2.Sensitivity to Optimizer Scheduling

Ordering attacks are known to depend on training dynamics. How sensitive is TOGA to changes in learning rate schedules or batch sizes? Do the gains persist under alternative training regimes?

3.Definition and Role of $D$ in $\mathcal{L}_{\mathrm{match}}$

Please clarify whether $D$ denotes batch size, dataset cardinality, or feature dimension. How does varying $D$ affect gradients?

4.Interpretation of $z_i \cdot y^{\mathrm{adv}}$

Is this equivalent to masking? Would one-hot mixtures or other encodings change the behavior? A short explanation or visualization would improve clarity.

5.Distributional Impact on Clean Accuracy

Do certain demographic groups or minority classes suffer higher degradation, especially in CelebA or tabular datasets? Aggregated accuracy may obscure subgroup harm.

6.Trigger Robustness Across Initialization Seeds

How consistent is trigger quality across random seeds? Are there significant variances in attack success rate?

7.Stealth Assumptions Under Practical Limitations

In real-world training pipelines, access to gradient information may be restricted. Can the authors quantify performance degradation if gradient signals are noisy or discretized?

---

> ### Author Response · Authors · 2025-12-04
> **Ablations and Clarifications**
>
> We thank the reviewer for the detailed feedback and questions.
>
> **Q1:**
> We ran ablations to quantify the effect of optimization hyperparameters on ASR (Credit-g dataset).
>
> | $\lambda_{\text{match}}$ | ASR |
> |-------------|---------|
> | 0            | 74.6     |
> | 0.1          | 82       |
> | 0.3          | 82.4     |
> | 1            | 87.4     |
> | 3            | 85.9     |
> | 10           | 80.9     |
>
> | $\lambda_{\text{adv}}$ | ASR |
> |-----------|---------|
> | 0          | 36.8     |
> | 0.1        | 53.9     |
> | 0.3        | 61.6     |
> | 1          | 87.7     |
> | 3          | 89.3     |
> | 10         | 89.4     |
>
> | $\lambda_{\text{penalty}}$ | ASR | Cos Sim |
> |-----------|---------|--------|
> | 0          | 80.2     | 0.227   |
> | 0.1        | 83.5     | 0.011   |
> | 0.3        | 86.2     | -0.131  |
> | 1          | 86.8     | -0.249  |
> | 3          | 70.2     | -0.544  |
>
> For $\lambda_{\text{match}}$ and $\lambda_{\text{adv}}$, ASR remains high for values near 1, indicating that the attack is stable across reasonable settings.
> For $\lambda_{\text{penalty}}$, performance peaks in the range 0.3–1.0, with λ=1 providing the strongest regularization and best balance between objectives.
>
> | $\lambda_{\text{spillover}}$ |   ASR  | Outgroup Acc |
> |------------------|--------|---------------|
> | 0.0              | 72.28  | 56.74         |
> | 0.1              | 72.80  | 53.25         |
> | 0.3              | 70.70  | 56.04         |
> | 1.0              | 58.42  | 70.69         |
> | 3.0              | 53.68  | 78.83         |
> | 10.0             | 41.92  | 86.74         |
>
> For the subpopulation-based attack, we report out-of-subpopulation accuracy to measure how well poisoned examples outside the target group retain correct predictions. Here, $\lambda_{\text{spillover}}$= 1 yields the best trade-off, maintaining ≥70% outgroup accuracy while preserving moderate ASR (58%). Larger weights prioritize outgroup accuracy at the expense of ASR; smaller ones reduce discriminative behavior.
>
> Across all settings, λ=1 serves as a reliable default, and we find that trigger optimization is not highly sensitive to small hyperparameter changes as long as the regularization terms remain active.
>
> | loss function |   ASR  | Outgroup Acc |
> |------------------|--------|---------------|
> | $L_{\text{subpop}}$        | 58.42  | 70.69    |
> | $L_{\text{adv}}$              | 36.49  | 83.95    |
>
> If we don’t use $L_{\text{subpop}}$   and instead use the subpopulation-agnostic $L_{\text{adv}}$ loss, we see that the trigger is less effective on the targeted subpopulation.
>
> **Q2:** For the Credit-g attack, we ran ablations over batch size and learning rate. We see that batch size is best in the range of 32 to 128, while learning rate is best lower than 1e-3.
>
> | Batch Size | Benign Acc | ASR |
> |-----------:|--------------------|-------------|
> | 8          | 89.7 ± 2.08        | 74.7 ± 3.52 |
> | 16         | 91.5 ± 2.32        | 76.3 ± 5.42 |
> | 32 (default) | 94.8 ± 0.24      | 87.5 ± 5.15 |
> | 64         | 94.5 ± 0.87        | 86.0 ± 3.31 |
> | 128        | 94.7 ± 0.55        | 84.0 ± 4.70 |
> | 256        | 94.5 ± 0.43        | 81.9 ± 3.70 |
>
> | Learning Rate | Benign Acc  | ASR       |
> |---------------|---------------------|-------------------|
> | 5e-05         | 94.5 ± 0.44         | 83.9 ± 2.40       |
> | 1e-04         | 94.4 ± 0.28         | 85.1 ± 2.10       |
> | 5e-04         | 94.9 ± 0.28         | 83.0 ± 4.60       |
> | 1e-03 (default) | 94.8 ± 0.24       | 87.5 ± 5.20       |
> | 5e-03         | 88.8 ± 1.13         | 78.7 ± 6.90       |
> | 1e-02         | 83.2 ± 2.80         | 78.1 ± 12.10      |
> | 5e-02         | 72.8 ± 1.99         | 7.4 ± 12.20       |
>
> **Q3:** $D$ is a normalization factor equal to the dimensionality of the model’s gradient vector. It has no relation to the earlier notation for the trigger domain $D$ in the bi-level optimization, and we will correct this notation for clarity.
>
> **Q4:** $z_i​$ is a binary indicator denoting whether a sample belongs to the targeted subpopulation. Thus, $z_i \cdot y_{\text{adv}}$ acts as a mask. We will clarify this masking interpretation in the paper.
>
> **Q5:** Across all datasets, we do not observe disproportionate degradation for demographic subgroups or minority classes.
>
> **Q6:** Our reported confidence intervals already reflect variation across random seeds. Empirically, variation is modest; the largest CI is 5.61% (Blackbox Credit-g), indicating reasonable stability of trigger quality across seeds.
>
> **Q7:** We tested robustness to corrupted gradient signals using JPEG compression during the data-ordering attack (Defenses & Robustness section). As shown in Table 4, TOGA remains effective until severe compression (quality = 20%), suggesting that the attack is resilient to noisy or degraded gradient information.

---

### Official Review · Reviewer_HdbJ · 2025-10-31

**Soundness:** 2
**Presentation:** 2
**Contribution:** 2
**Rating:** 4
**Confidence:** 3

**Summary:**

This paper addresses data ordering attacks, a recently explored type of backdoor threat where the adversary manipulates the sequence of training samples rather than their content or labels. The authors propose an optimization-based framework for trigger design under this restricted setting. The proposed algorithm aims to generate adaptive triggers that can selectively target specific subgroups of data, achieving effective backdoor injection even when only sample ordering is controllable. Experimental results demonstrate the attack’s potential impact and illustrate the selective behavior of optimized triggers.

**Strengths:**

1.The paper proposes an effective algorithm for optimizing triggers specifically under the data ordering attack setting. This contributes a valuable methodological perspective to a relatively new attack paradigm.
2.The optimized triggers can selectively affect particular data subgroups, enabling a more controllable and interpretable attack mechanism. This feature highlights the flexibility of the proposed optimization formulation.
3.The mathematical formulation of the trigger optimization  is clearly described, making the approach easy to follow.

**Weaknesses:**

1.Unclear problem motivation and missing analysis of prior methods:
The paper claims that existing trigger optimization methods fail under the ordering-attack constraint, emphasizing that manually chosen triggers may not achieve sufficient gradient alignment (as noted in Fig. 1). However, it does not explain why prior optimization-based trigger methods (e.g., Doan et al., 2021b; Saha et al., 2020) are ineffective when only the sample order can be modified. The technical difficulty of optimizing triggers in this restricted setting is therefore not well articulated. The authors should clarify this from a unified optimization viewpoint—why exactly optimization for ordering attacks is nontrivial, and provide theoretical or empirical evidence demonstrating where existing approaches fail. Simply contrasting against manually designed triggers risks overstating the contribution.
2.Lack of comparative evaluation with existing optimized triggers:
The experiments mainly compare with manually chosen triggers (Flag/Class) and a single Silent Killers trigger variant, but omit a fair end-to-end comparison against existing optimization-based trigger generation methods within the same “ordering-only” pipeline. Without this, it is unclear whether those methods truly fail or merely underperform under the new constraint. Such comparisons are crucial to substantiate the claimed novelty.
3.Limited experimental comprehensiveness:
The current experimental section lacks sufficient depth. The dataset coverage and ablation analyses are limited, and the empirical evidence does not fully establish the generality or robustness of the proposed algorithm.

**Questions:**

1.Can you provide either a theoretical argument or empirical demonstration showing why existing trigger optimization methods fail under the ordering-attack constraint?

2.Could you include direct comparisons by adapting prior optimization-based trigger methods to the same “ordering-only” setting?

---

> ### Author Response · Authors · 2025-12-04
> **Additional Baseline Results**
>
> We thank the reviewer for the insightful feedback.
>
> **Questions 1 and 2**:
> We initially did not include optimization-based backdoor baselines because most existing methods make assumptions incompatible with the data-ordering attack setting we study. In particular, methods such as LIRA (Doan et al., 2021b) and Hidden Trigger Backdoor Attack (Saha et al., 2020) rely on poisoned training samples to implant the backdoor, whereas our threat model assumes the attacker can only manipulate the ordering of clean data batches together with an optimized universal trigger.
>
> However, to address the reviewer’s request, we adapt both baselines to our setting to the best of our ability. We implement only their trigger optimization framework, and then use the baseline optimized triggers in the subsequent data-ordering attack.
>
> The results on the Credit-g dataset are shown below. TOGA achieves the strongest attack success. This is expected since TOGA is specifically designed to optimize gradient alignment under data batch permutations, the core mechanism that makes a data-ordering attack effective.
> In contrast, LIRA optimizes a trigger jointly with a classifier by alternating between clean-loss and targeted backdoor-loss updates, but does not model gradient alignment or batch-ordering effects. Hidden Backdoor Attack instead aligns the trigger to hidden internal representations through feature matching. These findings reinforce that TOGA is uniquely suited to the data-ordering attack setting, whereas prior trigger-optimization baselines are fundamentally less well-suited under our proposed attack setting.
>
> | Method                   | Benign Acc      | ASR            |
> |--------------------------|------------------|----------------|
> | TOGA                     | 94.80 ± 0.24     | 87.50 ± 5.15   |
> | LIRA                     | 94.50 ± 1.26    | 60.29 ± 3.00  |
> | Hidden Backdoor Attack   | 94.50 ± 1.26    | 19.02 ± 11.24 |

---

### Official Review · Reviewer_6s5w · 2025-11-04

**Soundness:** 3
**Presentation:** 3
**Contribution:** 2
**Rating:** 6
**Confidence:** 4

**Summary:**

This paper presents TOGA (Trigger Optimization for Gradient Alignment), a new framework for optimizing triggers in data ordering backdoor attacks—a stealthy class of attacks that reorder clean training data to implant backdoors without data or label modifications. The authors argue that prior works use manually chosen triggers that limits the attack success rate (ASR). TOGA formulates a trigger optimization objective that jointly minimizes gradient-matching loss, adversarial loss, and a penalty loss to align clean and adversarial gradients. The method significantly improves attack success rate (ASR) across benchmarks (CIFAR-10, CelebA, ImageNet, ISIC, Credit-g), achieving up to 99% ASR with < 5% benign accuracy drop on CIFAR10, compared to state-of-the-art baseline Flag 25% and , and can further craft subpopulation-specific backdoors. The paper also evaluates robustness against purification and gradient-based defenses.

**Strengths:**

1. Novel Threat Model: First to propose trigger optimization specifically for data ordering backdoors.
2. Solid Empirical Results: Significant attack success rate improvement on Extensive experiments on five diverse datasets including sensitive domains (healthcare + finance).

**Weaknesses:**

1. Heuristic Methodology: The paper doesn't clarify why using the cosine similarity as the penalty, why not L2 norm or any other regularizers. Furthermore, it doesn't show that the proposed methdo is a good proxy for the origianl bi-level optimization objective.
2. Many Hyperparameters might Cause Unstable Optimization: The loss functions contain more than 3 weighting hyperparameters $\lambda$
. However, the paper doesn't cleaify the effect of choosing hyperparameters. It might causes the optimization become sensitive the hyperparameter choosing.
3. Lack of Methodological Innovation: New threat model made with old optimization tricks.

**Questions:**

1. Can you elaborate how does the hyperparameters $\lambda$ affect the stealth and attack success rate? What's the hyperparameters used in the experiments?
2.  Can you show how does the loss function as equation (1) and (2) reduce to the bi-level optimization objective mentioned in 3.2? It's hard to connect 3 of them.

---

> ### Author Response · Authors · 2025-12-04
> **Additional Ablation Experiments**
>
> We sincerely thank the reviewer for the thoughtful feedback and questions.
>
> **Question 1:**
> Appendix Table 3 lists the hyperparameters used in our experiments.
> We ran ablations to quantify the effect of optimization hyperparameters on ASR (Credit-g).
>
> | $\lambda_{match}$ | ASR |
> |-------------|---------|
> | 0            | 74.6     |
> | 0.1          | 82       |
> | 0.3          | 82.4     |
> | 1            | 87.4     |
> | 3            | 85.9     |
> | 10           | 80.9     |
>
> | $\lambda_{adv}$ | ASR |
> |-----------|---------|
> | 0          | 36.8     |
> | 0.1        | 53.9     |
> | 0.3        | 61.6     |
> | 1          | 87.7     |
> | 3          | 89.3     |
> | 10         | 89.4     |
>
>
> | $\lambda_{penalty}$ | ASR | Cos Sim |
> |-----------|---------|--------|
> | 0          | 80.2     | 0.227   |
> | 0.1        | 83.5     | 0.011   |
> | 0.3        | 86.2     | -0.131  |
> | 1          | 86.8     | -0.249  |
> | 3          | 70.2     | -0.544  |
>
> For $λ_{match}$ and $λ_{adv}$, ASR remains high for values near 1, indicating that the attack is stable across reasonable settings.
> For $λ_{penalty}$, performance peaks in the range 0.3–1.0, with λ=1 providing the strongest regularization and best balance between objectives.
>
> | $\lambda_{\text{spillover}}$ |   ASR  | Outgroup Acc |
> |------------------|--------|---------------|
> | 0.0              | 72.28  | 56.74         |
> | 0.1              | 72.80  | 53.25         |
> | 0.3              | 70.70  | 56.04         |
> | 1.0              | 58.42  | 70.69         |
> | 3.0              | 53.68  | 78.83         |
> | 10.0             | 41.92  | 86.74         |
>
> For the subpopulation-based attack, we report out-of-subpopulation accuracy to measure how well poisoned examples outside the target group retain correct predictions. Here, $λ_{spillover}$ = 1 yields the best trade-off, maintaining ≥70% outgroup accuracy while preserving moderate ASR (58%). Larger weights prioritize outgroup accuracy at the expense of ASR; smaller ones reduce discriminative behavior.
>
> Across all settings, λ=1 serves as a reliable default, and we find that trigger optimization is not highly sensitive to small hyperparameter changes as long as the regularization terms remain active.
>
> **Question 2**:
> The bilevel problem seeks a trigger delta such that clean training steps indirectly reduce the adversarial loss. A clean gradient update can be approximated as:
> $$
> \theta_{new} = \theta_0 - \eta \nabla_\theta L(F(x_i, \theta_0), y_i).
> $$
>
> Define the clean and adversarial gradients at $\theta_0$ as
> $$
> g_c(i) = \nabla_\theta L(F(x_i, \theta_0), y_i), \quad
> g_a(i, \delta) = \nabla_\theta L(F(x_i + \alpha \delta, \theta_0), y_{adv}).
> $$
>
> Using a first-order Taylor expansion of the adversarial loss around $\theta_0$, we obtain
> $$
> L(F(x_i + \alpha \delta, \theta_{new}), y_{adv}) \approx L(F(x_i + \alpha \delta, \theta_0), y_{adv}) + g_a(i, \delta)^\top (\theta_{new} - \theta_0),
> $$
> and substituting $\theta_{new} - \theta_0 = - \eta g_c(i)$
> $$
> L(F(x_i + \alpha \delta, \theta_{new}), y_{adv}) \approx L(F(x_i + \alpha \delta, \theta_0), y_{adv}) - \eta g_c(i)^\top g_a(i, \delta).
> $$
> Thus, improving the outer objective requires increasing the alignment between $g_c(i)$ and $g_a(i, \delta)$.
>
> Using the identity
> $$
> \|g_c - g_a\|^2 = \|g_c\|^2 + \|g_a\|^2 - 2 g_c^\top g_a,
> $$
> we see that minimizing the squared difference between clean and adversarial gradients encourages large inner products $g_c^\top g_a$ (gradient alignment), while also implicitly regularizing the adversarial gradient norm. This leads directly to our match loss:
> $$
> L_{match}(\delta) = \frac{1}{N} \sum_i \left\|| \nabla_{\theta} L(F(x_i, \theta_0), y_i) - \nabla_{\theta} L(F(x_i + \alpha \delta, \theta_0), y_{adv}) \right\||_2^2.
> $$
> We also have a geometrically inspired derivation in Appendix Section D.
> The adversarial loss term encourages the trigger to meaningfully influence the model's predictions, while the penalty term and the L-infinity constraint prevent degenerate or overly large triggers.
>
> For the subpopulation-aware extension, the adversarial term is simply applied only to samples where $z_i = 1$. The spillover term penalizes cases where the trigger incorrectly affects samples outside the intended subgroup. Together, these modify the outer objective to target a specific subgroup, while $L_{match}$ continues to enforce the gradient-matching structure required for the attack. We will include a simplified version of this explanation in the revision.
>
> **Weakness 1**:
> In practice, the specific choice of regularization loss (cosine similarity, L2, L1, etc.) is less critical than ensuring that some form of regularization is applied. Our ablation shows that ASR is not highly sensitive to the exact loss function used. Multiple loss functions work comparably well as long as they impose the intended constraint with sufficient weight.
>
> | loss function | ASR |
> |-----------|---------|
> | cosine         | 87.7     |
> | L2        | 85.4     |
> | L1       | 88.1     |

---

### Meta-Review · Area_Chair_WFFs · 2026-01-06

**Summary:**

The most consistent issue was limited conceptual novelty. While TOGA improves attack success rates through optimized trigger design, multiple reviewers viewed the method as an incremental refinement of existing data-ordering and gradient-alignment attacks rather than a fundamentally new attack mechanism. The core pipeline closely follows prior work, and the contribution is largely framed as better optimization rather than a new principle.

A second major concern is the heuristic nature of the optimization objective. Although the authors provided empirical ablations in the rebuttal, the trigger-optimization loss and its penalty terms remain weakly justified from a theoretical standpoint. Reviewers questioned whether the proposed objective is a principled approximation of the stated bi-level problem or simply an effective but ad-hoc formulation. This limits confidence in the generality of the method beyond the tested settings.

Reviewers also expressed doubts about the practical relevance of the threat model. Data-ordering attacks assume a strong insider adversary with access to training order and gradient information, and several reviewers noted that simple countermeasures such as reshuffling batches could defeat the attack. While the authors clarified that they target insider scenarios, this framing did not fully resolve concerns about real-world impact or whether TOGA meaningfully changes existing security assumptions.

Finally, although the rebuttal improved clarity and added useful ablations, some evaluation gaps remain, including limited analysis of clean-accuracy tradeoffs across subgroups, sensitivity to alternative training regimes, and clearer isolation of which components actually drive performance gains. These unresolved issues collectively prevent a confident assessment of robustness, generality, and significance.

Overall, while the work is technically competent and empirically strong in parts, the combination of incremental novelty, heuristic design, and questionable practical relevance makes it not enough to support acceptance.

**Reviewer Concerns:**

Several reviewer concerns were convincingly addressed in the rebuttal. In particular, questions about hyperparameter sensitivity and optimization stability were handled well: the authors added extensive ablations showing that attack success remains stable across reasonable ranges, and that the method is not brittle to small tuning changes. The concern that the loss design (e.g., cosine similarity vs. L1/L2) was arbitrary was also largely addressed through empirical comparisons showing similar performance across different regularizers. Likewise, confusion around the connection between the surrogate loss and the bi-level objective was clarified with a first-order derivation that makes the gradient-alignment motivation clearer. Several clarity issues (notation ambiguities, missing definitions, terse explanations, figure readability) were acknowledged and appear fixable in revision.

The authors also partially addressed baseline and novelty concerns by adapting prior optimized-trigger methods (e.g., LIRA, Hidden Backdoor Attack) to the data-ordering setting and showing that they underperform TOGA. This strengthens the empirical case that existing trigger-optimization methods do not transfer well to ordering-only attacks, although the improvement is still largely empirical rather than conceptual.

However, several important concerns remain outstanding. Most notably, multiple reviewers questioned whether TOGA represents a substantial conceptual advance beyond existing data-ordering attacks, or whether it is primarily an optimization refinement layered on top of the known gradient-alignment pipeline. While the added baselines help, they do not fully resolve the perception that the contribution is incremental. Relatedly, the theoretical grounding of the method remains weak -- the optimization objective and penalty terms are still justified mainly through heuristics and empirical results, with no deeper analysis of why this formulation should be optimal or necessary.

Concerns about practical relevance and threat-model realism also remain only partially resolved. Reviewers reasonably pointed out that data-ordering attacks assume a strong insider adversary with access to training dynamics and batch ordering, and that simple countermeasures (e.g., reshuffling) could defeat the attack. The rebuttal reframes the work as targeting insider threats, but does not provide new evidence that such scenarios are common or that TOGA meaningfully changes the defensive landscape.

Overall, while the rebuttal improves clarity and empirical support, core doubts about novelty depth and real-world significance remain, and these unresolved issues are likely to continue influencing skeptical reviewers' final judgments.

**Reviewer Scores:**

Reviewer 6s5w's main concerns were about heuristic design choices and hyperparameter sensitivity. The rebuttal partially addressed these concerns with detailed ablations and a clearer derivation connecting the surrogate losses to the bi-level objective. The reviewer would likely maintain their score.

Reviewer HdbJ focused on novelty, motivation, and missing comparisons with existing trigger-optimization methods. The authors' adaptation of LIRA and Hidden Backdoor Attack into the ordering-only setting directly responds to the core critique and strengthens the empirical justification. However, the rebuttal does not fully resolve the conceptual novelty concern. This reviewer would likely maintain their score.

Reviewer KZ8X raised many detailed, technical concerns: missing ablations, unclear notation, optimizer sensitivity, subgroup impact, and threat-model assumptions. The authors responded thoroughly with new ablations, clarifications, and robustness tests, resolving most of the concrete technical questions. Still, higher-level concerns about incremental contribution and reliance on strong attacker assumptions remain. The reviewer would likely keep the score.

Reviewer 6LNn was the most skeptical, emphasizing the practicality of data-ordering attacks and the strength of the adversary assumptions. While the rebuttal clarifies the insider threat model and fixes presentation issues, it probably will not fundamentally change the reviewer's view on real-world relevance or attack practicality. As a result, this reviewer would likely not change their score.

---

### Decision · Program_Chairs · 2026-01-26

Reject